# Hydrogen Gas-Grilling in Meat: Impact on Odor Profile and Contents of Polycyclic Aromatic Hydrocarbons and Volatile Organic Compounds

**DOI:** 10.3390/foods13152443

**Published:** 2024-08-02

**Authors:** María José Beriain, Inmaculada Gómez, Susana García, José Carlos Urroz, Pedro María Diéguez, Francisco C. Ibañez

**Affiliations:** 1ISFOOD Research Institute, Public University of Navarre, 31006 Pamplona, Spain; pi@unavarra.es; 2Department of Biotechnology and Food Science, Faculty of Sciences, University of Burgos, 09001 Burgos, Spain; igbastida@ubu.es; 3Department of Sciences, Public University of Navarre, 31006 Pamplona, Spain; susana.garcia@unavarra.es; 4School of Industrial & ICT Engineering, Public University of Navarre, Campus de Arrosadía, E-31006 Pamplona, Spain; josec.urroz@unavarra.es (J.C.U.); pmde@unavarra.es (P.M.D.)

**Keywords:** alternative fuel, thermal treatment, odor compounds, PAHs, safety

## Abstract

The effect of fuel (hydrogen vs. butane) on the formation of volatile organic compounds (VOCs) and polycyclic aromatic hydrocarbons (PAHs) was evaluated for grilled horse meat (very low-fat and low-fat) cooking vertically. Gas chromatography-mass spectrometry was used to analyze PAHs and VOCs. An electronic nose was used to evaluate the odor profile. Total high-molecular-weight PAHs ranged from 19.59 to 28.65 µg/kg with butane and from 1.83 to 1.61 µg/kg with hydrogen. Conversely, total low-molecular-weight PAHs went from 184.41 to 286.03 µg/kg with butane and from 36.88 to 41.63 µg/kg with hydrogen. Aldehydes and alkanes were the predominant family in a total of 59 VOCs. Hydrogen gas-grilling reduced significantly (*p* < 0.05) the generation of VOCs related to lipid oxidation. The odor profile was not modified significantly despite the change of PAHs and VOCs. The findings indicate that hydrogen is a viable alternative to butane for grilling horse meat. Hydrogen gas-grilling may be regarded as a safe cooking procedure of meat from a PAH contamination point and perhaps sustainable environmentally compared to a conventional technique. The present study provides the basis for the use of hydrogen gas in grilled meat.

## 1. Introduction

Meat can be cooked using various methods, such as steaming, frying, roasting, or grilling, to enhance the formation of volatile organic compounds (VOCs) [1]. However, these cooking methods can also produce harmful compounds, such as polycyclic aromatic hydrocarbons (PAHs), particularly when grilling with solid (charcoal) or gaseous (butane/propane) fuels. In addition, the PAH content in beef is not solely dependent on the type of fuel utilized but also on the proximity to the heat source and direct contact with the flame [2]. PAHs comprise a class of organic compounds of petrogenic origins or generated from the incomplete combustion of organic matter, which takes place during food processing by heat treatments such as smoking meat products or cooking meat above 150 °C [3]. The Joint FAO/WHO Expert Committee on Food Additives and the Scientific Committee on Food (SCF) consider PAHs to be genotoxic and carcinogenic and recommend monitoring the presence of PAHs in food [4]. Several lines of evidence indicate that cooking conditions and dietary habits can contribute to human cancer risk through the ingestion of genotoxic compounds from heat-processed foods [5]. European legislation restricts the presence of benzo[*a*]pyrene to 5 µg/kg and the sum of the four PAHs benzo[*a*]pyrene, benzo[*a*]anthracene, benzo[*b*]fluoranthene, and chrysene (PAH4) to 30 µg/kg in smoked meat and smoked meat products [6]. However, these regulations allow the limits to be exceeded in some foodstuffs produced by certain countries (Ireland, Croatia, Cyprus, Spain, Poland, Portugal, Latvia, Slovakia, Finland, and Sweden). Several strategies have been suggested to reduce the formation of PAHs in meat products by modifying cooking conditions, such as reducing grilling temperature and time or adding antioxidants to foodstuff during grilling [7]. The use of marinates with acidic pH and enriched in phenol compounds and the use of synthetic casings in smoked products could be useful strategies to decrease the PAH content [8]. Furthermore, the technique of cooking the meat can affect the PAH formation [9]. Saint-Aubert et al. [10] reported that cooking meat horizontally with oil dripping directly into the fire source results in more PAH formation than cooking it vertically.

The scent of cooked meat is influenced by factors such as meat aging (proteolysis of myofibrillar protein and lipolysis of intramuscular fat) and cooking methods (stewing, roasting, frying, or grilling) [1]. During the cooking process, various chemical reactions arise. These reactions occur at temperatures above 150 °C. A fast and high-temperature cooking method produces more Maillard reaction products, whereas a slow and low-temperature cooking method mainly produces lipid degradation products [11]. These reactions occur simultaneously, resulting in the formation of VOCs such as aldehydes, ketones, alcohols, hydrocarbons, pyrazine derivatives, and sulfur compounds. These compounds create a unique odor profile for each type of meat and cooking method. The odor of grilled meat is usually attributed to ketone products derived from lipids. The major ketones reported in grilled horse meat were acetophenone, 2-pentanone, and 2-heptanone [3]. The human nose can assess the VOCs generated by meat products, which are related to their quality. However, the use of a trained panel is costly and time-consuming. The electronic nose system can detect and differentiate among some odoriferous compounds in a fast and cost-effective manner [12]. In contrast to traditional analytical techniques, e-nose technology does not focus on identifying and quantifying individual components of a volatile compound mixture. Instead, it emphasizes the quantitative description of the entire aroma profile, encompassing the interrelationships among its components [13].

The European countries with the highest annual consumption of horse meat are Belgium (1.2 kg/person), Italy (1.0 kg/person), the Netherlands (1.0 kg/person), and Luxembourg (0.75 kg/person) [14] To date, studies have been conducted on the levels of PAHs in grilled beef, pork, and chicken [15]. However, no research has been performed on the levels of these compounds in horse meat or fuel substitution to mitigate their formation. As for the research on VOCs in horse meat, all of them used an electric grill [16,17,18,19]. No studies have evaluated the impact of gas-grilling vertically on VOCs in horse meat. Finally, the e-nose has been used to examine the aroma of grilled chicken [20] and lamb [21] cooked with charcoal as fuel, as well as beef cooked on an electric grill [22]. No studies have been found where an e-nose was used to analyze the odor profile of horse meat cooked on a grill using gaseous fuel. The present study hypothesized that vertically grilling horse meat using hydrogen fuel could reduce PAH content and preserve the profile of VOCs. The objective of the present research was to quantify the impact of fuel type (butane vs. hydrogen gas) on the formation of PAHs and VOCs and odor profile in grilled horse meat cooked vertically with variable low-fat content.

## 2. Materials and Methods

### 2.1. Materials

#### 2.1.1. Meat Sample

The meat samples (40 kg of *Longissimus dorsi lumborum* of the Burguete breed, Navarre, Spain) were provided by a local supplier and transported to the laboratory under refrigeration. A total of 80 steaks (350 g and 2.5 cm thick each) were prepared. Forty steaks were grilled with butane and the other forty with hydrogen gas. The proximal composition of the meat samples was established according to standardized norms (Appendix A). Samples with less than 1% fat (as a cut-off) were classified as “very low-fat” (VLF, *n* = 40), while those with more than 1% were categorized as “low-fat” (LF, *n* = 40). In the present study, lean horse meat was selected because it is known that the level and type of fat influence the formation of volatile compounds and PAHs. Therefore, in this study, low-fat horse meat was chosen to reduce external factors that may affect the formation of volatiles and PAHs, thus allowing us to focus on the effect of the type of fuel used in grilling.

#### 2.1.2. Reagents and Standards

A mix of 16 PAH standards (AccuStandard Inc., New Haven, CT, USA) with a concentration of 0.2 mg/mL in dichloromethane/methanol (50:50) was used. In addition, sixteen separate PAHs (Scharlab SL, Madrid, Spain) as standards were used. A mixture of deuterated PAHs (AccuStandard Inc., New Haven, CT, USA) including d_12_-chrysene, d_10_-phenanthrene, d_8_-naphthalene, d_12_-perylene, and d_12_-benzo[*a*]pyrene, was used at a concentration of 100 µg/mL in dichloromethane. In addition, d_10_-acenaphthen- was purchased from Dr. Ehrenstorfer (LGC Ltd., Middlessex, UK) and was used at 10 µg/mL. A solution of deuterated standards in cyclohexane/ethyl acetate (9:1) with a concentration of 500 µg/kg was prepared.

The PAH extraction system from the sample was Bond Elut QuEChERS kits (QuEChERS extraction kit, 5982-5650CH and QuEChERS dispersive kit, 5982-5156CH) from Agilent Technologies Inc. (Santa Clara, CA, USA). The solvents used (cyclohexane, ethyl acetate, and acetonitrile) were all chromatographic grade (MS SupraSolv^®^ from Merck Life Science SLU, Madrid, Spain).

### 2.2. Sample Preparation and Gas-Grilling Method

One steak with each fuel was cooked on a double stainless-steel grill (measuring 40 × 45 cm) using butane and hydrogen gas. To achieve a heating power comparable to that of a domestic cooker burner (about 1.4 kW), and taking into account the lower heating values and molecular masses of the gases, the hydrogen and butane volumetric flows were 0.463 and 0.042 Nm^3^/h, respectively (F-112AX-HEE-99-V gas flowmeter, Bronkhorst High-Tech B.V., Ruurlo, The Netherlands). Both fuels were pre-mixed with stoichiometric air. A total of 20 cooking sessions were conducted, with the steaks positioned vertically (Figure 1) to prevent the fat from dripping into the burner nozzle. During grilling, the internal temperature of the meat samples was measured using a digital thermometer with a platinum Pt100 probe (HD 9219 Delta Ohm SRL, Selvazzano, Italy). The temperature in the sample center reached 70 °C. To monitor the temperature at the surface, an infrared vision thermographic camera (testo 872s, Instrumentos Testo SA, Barcelona, Spain) was used. The torch was applied for 2 min until the sample surface of samples reached approximately 200 °C. The grill was cleaned after each cooking. After cooking, each sample was split into three portions for analysis of PAHs, VOCs, and odor profile. Following cooling, all samples were vacuum-sealed and stored at −20 °C until analysis according to the method by Cittadini et al. [23]. Before analysis, each sample was thawed at 4 °C for 24 h and homogenized using a domestic grinder (Moulinette 1,2,3-Moulinex, Grupo SEB Ibérica S.A. Barcelona, Spain).

### 2.3. Polycyclic Aromatic Hydrocarbon Analysis

#### 2.3.1. Extraction by QuEChERS Method

The QuEChERS method adapted to meat products was used [24]. A total of 150 µL of the internal standard and 15 mL of acetonitrile were added to a 50 mL centrifuge tube containing 3 g of sample. The mixture was vortexed using a Reax 2000 (Heidolph Instruments GmbH & Co. KG, Schwabach, Germany), and a ceramic stick was added. The QuEChERS extraction kit was then added, and the mixture was shaken again and centrifuged at 4000 rpm for 5 min (5910 Ri, Eppendorf Ibérica SLU, Madrid, Spain). The sample was prepared by transferring 9 mL of the supernatant to a 15 mL centrifuge tube and left to stand at −20 °C for 30 min. Then, 6 mL of the supernatant was added to a tube containing the dispersed phase of the QuEChERS kit, along with an additional 0.3 g of C18 and a ceramic rod. The mixture was shaken and centrifuged at 4000 rpm for 5 min. Next, an aliquot of the supernatant was left overnight at 4 °C. Finally, 1.5 mL of the extract was poured into a chromatography vial. The sample was evaporated using a nitrogen stream and then reconstituted with 300 µL of cyclohexane/ethyl acetate (9:1) before being filtered with a 22 µm filter and injected into the chromatograph system.

#### 2.3.2. GC-MS Analysis

The analysis was performed using an Agilent 7890B chromatographic system (Agilent Technologies Inc., Santa Clara, CA, USA) equipped with an autosampler (PAL3 RSI 120), a mass selective detector (Agilent 5977), and a J&W Select PAH GC column (30 m, 0.25 mm, 0.15 µm). A sample volume of 1.5 µL was injected in pulsed splitless mode (injector temperature 300 °C and constant flow rate of 1 °C/min), and the method reference was Duedahl-Olesen et al. [25]. The oven program was as follows: 60 °C for one minute, then increase at 25 °C/min until reaching 180 °C, then increase at 7 °C/min until reaching 298 °C, then increase at 2 °C/min until reaching the final temperature of 325 °C, and hold at that temperature for one minute. Following the run at 325 °C for 2.8 min, mass spectroscopy conditions were as follows: transfer line temperature 325 °C, source temperature 230 °C, and quad temperature 150 °C. The Agilent MassHunter Integrated GC Software (Version 10.1) was employed for the data acquisition method, utilizing Selective Ion Monitoring (SIM) mode with 8 segments (Appendix A).

#### 2.3.3. Calibration, Quantification, and Method of Validation

A calibration curve was performed for each of the 16 PAHs at concentrations of 1, 2.5, 5, 10, 25, and 50 ng/mL of the analyte and 25 ng/mL of its corresponding internal standard, using cyclohexane/ethyl acetate mixture (9:1) as solvent. Retention times were established by comparison with the individually injected standards (Appendix A). The concentration of each PAH in the sample was calculated based on the calibration curve and considering the response factor of the corresponding internal standard. The data were expressed on wet matter.

An analytical control (validation) of the method was conducted by adding known quantities of the various PAHs and internal standards to the same matrix (fortification), using the sample with the lowest concentration of PAHs and calculating the recovery percentage. This was performed at the lowest (1 ng/mL), middle (5 ng/mL), and highest (50 ng/mL) levels of the present working range. PAHs used in this study and their absolute recoveries and relative standard deviations are summarized in Appendix A.

### 2.4. Volatile Organic Compounds by GC-MS Analysis

For the analysis of volatile organic compounds (VOCs), the method described by Gorraiz et al. [26] was used. Approximately 10 g of thawed and homogenized samples were placed in a headspace vial. The sample was purged with a 40 mL/min helium flow for 10 min at 70 °C using a 4460th purge-and-trap concentrator (OI Analytical Instrument Management, Littleton, CO, USA). The headspace VOCs were collected in a Tenax GC trap (60/80 mesh) at 30 °C and thermally desorbed at 180 °C for 4 min with a 40 mL/min helium flow rate. A Hewlett-Packard HP-5890 gas chromatograph equipped with an HP-5 capillary column and controlled by OpenLab CDS software (Version 2.2) was used. Helium was used as the carrier gas to perform the analysis. The column was heated to 35 °C for 15 min and then programmed to reach 220 °C with a ramp of 8 °C/min, which was maintained for 5 min. Duplicate determinations were made for each sample. VOCs were identified using an HP-6890 gas chromatograph (Hewlett-Packard Co., Madrid, Spain) coupled with an HP-5973 quadruple mass spectrometer (Hewlett-Packard Co., Madrid, Spain).

The operational parameters were as follows: initial voltage, 70 eV; ion source temperature, 230 °C; electron multiplier voltage, 3000 V; scan speed, 3.32 scan/s; and scan range, 30 to 250 uma, with the quadrupole at 108 °C. The spectra obtained were compared with the Wiley 275 GC/MS library for Volatile Compounds in Food (Wiley, New York, NY, USA). The retention rates were calculated using the Castello et al. [27] method, modified for temperature-programmed gas chromatographs.

### 2.5. Odor Profile by e-Nose

The odor profile was measured directly on the ground meat samples. Each analysis was performed 5 times for each sample. The method followed was the one described by Doleman et al. [28], with some modifications. The extraction of volatile compounds was performed by static head space. In short, 2 g of ground meat was introduced into vials and incubated for 10 min at 55 °C with an autosampler (HS100 CTC-Combi-Pal, CTC Analytics AG, Zwingen, Switzerland) to determine the odor profile of each type of cooked meat. The extracted volatile fraction (a volume of 1.5 mL of the headspace) was injected into the Alpha MOS electronic nose, model α-FOX 4000 (Alfa MOS, Toulouse, France), equipped with 18 metal oxide sensors. Analysis of the sensor responses collected for two minutes was performed. The results obtained were processed to obtain the fingerprints of the volatile organic compounds (VOCs) present in each analyzed cooked meat sample. The resistance in each sensor was assayed by the software AlphaSoft version 9.1 (Alpha Software Corporation, Burlington, MA, USA), and the response intensity was calculated as previously described [28].

### 2.6. Statistical Analysis

Data were assessed by descriptive and inference analysis according to procedures described by Devore [29]. All the data were subjected to descriptive and inferential statistics. The parameters analyzed were evaluated by two-way ANOVA (fuel type and fat content). The interaction term was considered. Tuckey’s test was used for multiple comparisons and Student’s *t*-test was used for the comparison of two means. A Wilkis λ-based discriminant analysis was applied to PAHs and volatile compounds to assess the impact of the fuel type and fat content. All analyses were carried out with the IBM SPSS statistical package 28.0 version (IBM Corp., Armonk, NY, USA).

## 3. Results and Discussion

Appendix A summarizes the proximal composition of the meat. Moisture, protein, and ash values were higher than those reported by other authors [30,31,32] with little dispersion. In contrast, fat contents ranged from 0.23 to 3.38%, which is lower than the values reported by the previous authors.

### 3.1. Effects of Gas-Grilling on PAHs

In this study, four different horse meat samples, according to the fuel type and fat content, were evaluated. Table 1 displays the PAH contents of the gas-grilled samples with butane and hydrogen. It is evident that both fuels produced more low-molecular-weight (LMW) PAHs than high-molecular-weight (HMW) PAHs. The total LMW PAHs ranged from 184.41 ± 77.73 to 286.03 ± 95.66 µg/kg in butane gas-grilled samples and 36.88 ± 5.56 to 41.63 ± 11.47 µg/kg in hydrogen gas-grilled samples. In contrast, the total HMW PAHs ranged from 19.59 ± 13.36 to 28.65 ± 9.70 µg/kg in butane gas-grilled samples and 1.83 ± 0.38 to 1.61 ± 0.57 µg/kg when hydrogen gas was used in the grilling. Therefore, the use of gas for grilling, compared to the use of butane, reduced the total amount of PAHs, both LMW and HMW (*p* < 0.05).

Regardless of the fat content, the most abundant LMW PAHs of the butane gas-grilled samples were acenaphthylene (43.45–73.34 µg/kg) followed by naphthalene (41.62–68.63 µg/kg), pyrene (34.56–51.13 µg/kg), phenanthrene (26.72–39.01 µg/kg), and fluoranthene (17.82–27.00 µg/kg), whereas benzo[*ghi*]perylene (9.61–13.97 µg/kg) was the most abundant HMW PAH. Moreover, the hydrogen gas-grilling significantly reduced (*p* < 0.05) the PAHs in the horse meat, with naphthalene (23.43–28.36 µg/kg), acenaphthene (6.06–7.22 µg/kg) and phenanthrene (2.68–3.24 µg/kg) being the most abundant PAHs. The presence of only two benzene rings causes naphthalene to be more prone to formation than other PAHs [33].

In the butane gas-grilled samples, the PAH content was significantly higher (*p* < 0.05) when the fat content was higher. Fats play an important role in the formation of PAHs since lipophilic components are the major precursors of PAHs [33]. Furthermore, it has been observed that a high moisture content in meat and meat products can decrease the formation of PAHs by limiting the temperature increase during cooking [8]. The fact that the water content of VLF horse meat in the present study is higher than that of LF horse meat (Appendix A) supports the lower PAH content of VLF horse meat compared to LF horse meat butane gas-grilled.

The hydrogen gas-grilled samples of the present study did not significantly alter (*p* > 0.05) their PAH content when the fat content varied (Table 1). In addition to this, discriminant analysis was employed to classify the gas-grilled samples. Two functions were generated by the analysis. Discriminant function F1 explained 91% of the variance, while the discriminant function F2 explained 6.4% of the variance. As shown in Figure 2, samples cooked with hydrogen gas were in the same quadrant, suggesting that the use of hydrogen gas in grilling leads to a similar profile of PAHs in horse meat, regardless of their fat content. In contrast, the samples grilled with butane gas appear to be more widely dispersed and separated by the discriminant function 2, which was positively associated with naphthalene. Based on these results, the higher fat content could not be considered the primary factor contributing to the higher PAH levels observed in the VL butane gas-grilled horse meat. Similar results were obtained by Aaslyng et al.[15], who studied the PAH content in meat (pork, chicken, and beef) when barbecued. The analysis of PAH showed a markedly higher concentration of PAH in beef compared with chicken, which had a lower fat content than beef. Despite their results, these authors reported that the higher fat content in beef could not be the main contributor to the higher BaP and PAH contents due to the fat being intramuscular. It should be considered that most of the fat in the horse meat used in this study was intramuscular fat, which does not normally melt during cooking [15]. The same authors concluded that as the cooking time and temperature increase, the surface of the meat becomes darker, which correlates with the increase in PAH content. In the present work, the grilled meat showed optimal characteristics of appearance (slightly browned), especially those cooked with hydrogen gas.

The levels of PAH formation in thermally processed meat are influenced by the meat type, processing/cooking method employed, temperature, distance between the food and the heat source, duration and type of fuels, food additives used, etc. [34]. These factors probably reflect the chemical complexity of the generation of PAHs. When meat is grilled over high temperatures, PAHs are produced through the condensation of nitrogen-containing smaller organic compounds (such as amino acids and proteins) via pyrolysis at elevated temperatures. The precise mechanism of PAH production remains unclear, and it is hypothesized that their formation also occurs through free radical reactions, cyclization, and intramolecular addition of small molecules (pyrosynthesis) [7]. Likewise, the following chemical reactions and mechanisms have been proposed for the formation of PAHs: Bitter-Howard mechanism (addition of phenyl and benzyl radicals to yield naphthalene and methylnaphthalene, respectively), Badger mechanism (dehydrogenation reactions by pyrolysis in the intermediate product generator leading to formation of 3,4-benzopyrene), Frenklach mechanism (formation of PAHS from benzene rings by a dual pathway involving phenyl and naphthalene radicals), C_5_H_5_ mechanism (decomposition of benzene to cyclopentadienyl radicals, which combine to form naphthalene), Maillard reaction mechanism (pyrolysis of the Amadori rearrangement product forms 5-hydroxymethylfurfural and furfural, both of which yield PAHs), Diels-Alder reaction pathways (cyclization reaction in which a dienophile and a conjugated diene are joined to form a cyclic product by the formation of carbon-carbon bonds), and lipid oxidation pathway (reactions in which triacylglycerides undergo decomposition, dehydrogenation, hydrolysis and oxidation processes) [35]. The present study demonstrates that these reactions are modified when hydrogen is used, and further research would be interesting to understand the mechanisms of these changes when hydrogen gas is used for grilling meat.

Hydrogen gas-grilling significantly reduced (*p* < 0.05) PAHs, except for the PAH acenaphthene (Table 1). Likewise, samples were separated according to the fuel used (butane vs. hydrogen) in the discriminant analysis (Figure 2) by the discriminant function F1, which, as previously reported, explained 91% of the variance and was negatively associated with pyrene, phenanthrene, fluoranthene, and chrysene. Therefore, it could be speculated that the production of PAHs is lower when hydrogen gas is used in the grilling of the horse meat. The reduction of the PAHs implies a decrease in the potential carcinogenic compounds such as chrysene, benzo[*a*]anthracene, benzo[*ghi*]perylene, indeno[1,2,3-*cd*]pyrene, benzo[*a*]pyrene (BaP), benzo[*b*]fluoranthene, benzo[*k*]fluoranthene, and dibenzo[*a,h*]anthracene. These eight PAHs, either individually or collectively, are the sole viable indicators of the carcinogenic potential of PAHs in food [36]. It should be highlighted that the contents of PAHs of the samples of the present study were lower than 5 µg/kg for benzo[*a*]pyrene and 30 µg/kg for PAH4 (benzo[*a*]anthracene, benzo[*b*]fluoranthene, BaP, and chrysene), which are the limits in smoked meat and smoked meat products [6]. Even though the cooking method used in this study differs significantly from smoking, and since no specific limits are proposed by the EC (European Commission) for other meat products, assessments were conducted by comparing the outcomes with the EC limit. Since detection levels were below the EC legal limits, both butane and hydrogen gas-grilling could be deemed safe from PAH contamination in horse meat with low-fat content (lower than 1.9%). However, it should be highlighted that in the present study, the fat content of samples was low. Therefore, more studies with fattier meats and different species would be necessary to draw conclusions about PAH contamination in grilled meats.

### 3.2. Effects of Gas-Grilling on VOCs

The influence of gas-grilling and fat content on the formation of VOCs of cooked horse meat (expressed as AU × 10^6^/g dry matter) is summarized in Table 2. Fifty-nine out of the ninety-one volatile organic compounds (VOCs) identified were quantifiable and classified into seven families (aldehydes, alcohols, ketones, alkanes, aromatic hydrocarbons, sulfur-containing compounds, and other compounds). The total number of compounds detected varied slightly compared to those found in previous samples of horse meat [23,37]. The most prevalent compounds were aldehydes, alkanes, and alcohols, while furans and other compounds were present in smaller amounts.

Regarding the fat content, there were no significant differences (*p* > 0.05) in the butane gas-grilled horse meat, except for the ethanal, 2-pentanone, and 2,4-octadiene, in which the content was lower in LF samples (Table 2). In the case of horse meat grilled with hydrogen gas, there were no significant differences (*p* > 0.05) in fat content, except for ethanal, octanal, butanal, 3-methylbutanal, 3-ethyl heptane, and ethylbenzene, in which the content of VOCs was lower in samples with LF level. Moreover, some VOCs such as butanal, 2-penten-1-ol, 2-pentanone, 1-penten-3-one, 3-hexanone, heptane, hexane, 3-ethyl-1,5- octadiene, and ethenylbenzene were higher in LF hydrogen gas-grilled samples compared with their VLF counterparts. Thus, a higher fat content could increase the formation of certain volatile compounds due to the thermal decomposition of lipids and their interaction with other components of the meat [3]. However, further research would be needed to precisely confirm this hypothesis for grilled horse meat cooked with hydrogen gas. In addition to this, as previously performed for the PAHs, a discriminant analysis was employed to classify the gas-grilled samples according to their VOCs. Two functions were generated by the analysis. Discriminant function F1 explained 99% of the variance, while the discriminant function F2 explained 0.9% of the variance. As shown in Figure 3, samples cooked with butane gas were grouped, whereas the samples grilled with hydrogen gas were more widely dispersed and separated by the discriminant function 2 according to the fat content. However, it should be highlighted that this discriminant function 2 explained only 0.9% of the variance. The obtained results could suggest that the use of butane or hydrogen gas in a grill leads to a similar profile of VOCs in horse meat, regardless of their fat content. However, Bassam et al. [3] reported that the formation of volatile organic compounds (VOCs) in grilled meat was influenced by several factors, such as the native fat content. Likewise, Bleicher et al. [1] proposed that increasing the fat content and the amount of saturated fatty acids may result in an increase in the formation of VOCs, which in turn may lead to an intensification of the meat flavor. Nevertheless, it should be considered that in the present study, the fat level of the VLF samples was 0.51% and 1.85% for the LF samples (Appendix A). This reduced difference in fat level could explain why the fat amount was not a determining factor in the quantity of VOCs in the gas-grilled samples of the present study. However, it should be noted that if meat with medium or high-fat content had been used, there could have been different results, and the VOC contents might have shown another significant variation.

Hydrogen gas-grilled meat samples generally had lower VOC levels than butane gas-grilled samples, except for three alcohols, three ketones, and two alkanes. Although most VOCs had lower levels in hydrogen gas-grilled samples, the difference for the total VOCs was only 6 AU × 10^6^/g dry matter between the samples according to the fuel used. It should be highlighted the hexanal and (*Z*)-2-octene due to its high concentration in the samples and the effect of the gas used on their concentrations. While the hexanal was lower when the hydrogen gas was used (butane: 20.03–21.42; hydrogen: 0.84–0.37), the (*Z*)-2-octene was higher (butane: 0.04–0.06; hydrogen: 13.77–28.71).

The most abundant chemical family detected in cooked horse meat in the headspace were aldehydes and alkanes. The aldehydes ranged from 27.37–36.84 AU × 10^6^/g dry matter in hydrogen gas-grilled samples to 58.13–59.76 AU × 10^6^/g dry matter in samples grilled with butane. The volatile compounds found in cooked samples that are likely to have been generated from lipid oxidation include pentanal, hexanal, 2-hexenal, heptanal, benzaldehyde, octanal, 2-octenal, nonanal, and 3-ethyl benzaldehyde [18]. Aldehydes and hexanals are representative of the degree of lipid oxidation in cooked foal steaks [17]. The reduction of the aldehydes with the use of the hydrogen was mainly due to the hexanal, whose concentration was reduced from 20.03–21.42 (butane gas-grilled samples) to 0.84–0.37 AU × 10^6^/g dry matter (hydrogen gas-grilled samples). The results of the present study suggest that hydrogen grilling could significantly decrease lipid oxidation, which occurs for cooking treatments in foal meat [18]. The moderate oxidation of fat plays an important role in the formation of the characteristic cooked foal flavor [16]. Likewise, aldehydes are, in general, the major contributors to cooked horse meat aroma [37].

Moreover, the second abundant chemical family detected in gas-grilled horse meat in the headspace was alkanes, which ranged from 30.78–47.75 AU × 10^6^/g dry matter in hydrogen gas-grilled samples to 20.81–27.77 AU × 10^6^/g dry matter in samples grilled with butane. The increase of the alkanes with the use of hydrogen was mainly due to the (*Z*)-2-octene, whose concentration was increased from 0.04–0.06 (butane gas-grilled samples) to 13.77–28.71 AU × 10^6^/g dry matter (hydrogen gas-grilled samples). (*Z*)-2-octene is an aliphatic hydrocarbon whose origin can be attributed to lipid oxidation and which contributes to the pleasant flavor of grilled meat [3].

Therefore, as explained previously, the composition of each VOC is different according to the fuel used (Table 2). Likewise, samples were separated according to the fuel used (butane vs. hydrogen) in the discriminant analysis by the discriminant function F1 (Figure 3), which as previously reported, explained 99% of the variance and was strongly associated with compounds such as hexanal. The formation of VOCs is related to several factors, such as the temperature reached by samples during cooking or the fat of samples in foal meat [17]. However, according to the obtained results, it could be hypothesized that the production of VOCs differs according to the gas used in the grilling, and when the hydrogen gas was used, the VOC profile improved due to the reduction of aldehydes, which primarily originate from lipid oxidation

### 3.3. Effects of Gas-Grilling on Odor Profile by e-Nose

The odor profile resulting from the influence of PAHs and VOCs was evaluated by e-nose to determine its practical impact on the quality of VLF and LF horse meat grilled with hydrogen or butane. Appendix A summarizes the responses of the 18 sensors used in the e-nose system. Five sensors (LY2/Gh, LY2/gCTI, LY2/gCT, P30/2, and T40/2) did not detect differences among the samples. The remaining sensors recorded the lowest responses in the VLF samples grilled with butane.

A multivariate analysis of the response of sensors of e-nose was performed in order to classify the samples according to their fat content and the fuel used in the grilling of horse meat (Figure 4). Two functions were generated by the analysis. Discriminant function F1 explained 57.9% of the variance, whereas the discriminant function F2 explained 31.5%. As shown in Figure 4, samples grilled with hydrogen gas were grouped, whereas the samples grilled with butane gas were more widely dispersed and separated by the discriminant function 2 according to the fat content. The obtained results could suggest that the fat level of the gas-grilled horse meat does not influence the odor profile regardless of the gas used for the grilling. As previously discussed for PAHs and VOCs, the fat level was a determining factor for the composition of PAHs and VOCs, and this, in turn, was also reflected in the results of the electronic nose, resulting in a similar odor profile among the samples regardless of their fat level.

In contrast, samples were separated according to the fuel used (butane vs. hydrogen) in the discriminant analysis by the discriminant function F1 (Figure 4). These data corroborate those obtained for PAHs and VOCs, which were indeed affected by the type of gas used on the grill. However, the percentage of variance explained by Function 1 in the case of the electronic nose (57.59%) was lower than the variance explained by Function 1 in the PAHs (91%, Figure 2) and VOCs (99%, Figure 3). This suggests that the discrimination of samples by the electronic nose is lower than with the other techniques. This could be explained by the VOCs and PAHs that were more affected by using different gases on the grill. For instance, in the case of VOCs, hexanal and (*Z*)-2-octene were the most abundant compounds that were significantly affected by the fuel type and could be the compounds responsible for the change in the odor profile in the samples when hydrogen gas is used in the grilling of horse meat.

The obtained results in the present study highlight that compounds significantly present in the profile of VOCs and PAHs may not be perceptible by the human sense of smell. In addition to this, there is not always a straightforward relationship between the concentration of specific compounds and the perception of odor [38]. The e-nose technique is not concerned with identifying and quantifying the individual components of a mixture of volatile compounds. Rather, it is a method for describing the complete aromatic profile and the interrelationships between the compounds [39]. Therefore, using hydrogen as fuel in the horse meat grill would improve the composition of PAHs and VOCs in the grilled horse meat without significantly affecting its odor profile.

## 4. Conclusions

From the present study, it can be concluded that the fuel type in grilling and fat level affected the polycyclic aromatic hydrocarbons (PAHs) content. Hydrogen gas-grilling decreased the PAH content compared with butane gas-grilling, which would lead to healthier cooked meat. Moreover, whereas the higher the fat level, the higher the PAH content in butane gas-grilled horse meat, the fat level did not affect the PAH content when hydrogen gas-grilling was used. Thus, fuel type seemed to be the more important factor compared with fat level for PAHs. Regarding the volatile organic compounds (VOCs), regardless of the fat level, hydrogen gas-grilling led to a decrease in most VOCs, including those indicative of lipid oxidation. Thus, the type of gas used in grilling played a pivotal role in this study. Despite the change in the composition of PAHs and VOCs, the odor profile of the samples was not modified significantly. The e-nose simulates the human nose, evaluating the olfactory profile of a sample holistically. If the e-nose does not differentiate between the odor profile of the samples, neither would the consumer’s nose. Therefore, the change of fuel (hydrogen by butane) used in the grilling of horse meat improved the composition of VOCs and PAHs but did not affect the odor profile. In other words, consumers would benefit from the improved chemical composition of the samples with the new fuel used in the grilling without adversely compromising their sensory odor profile. From the perspective of PAH contamination, grilling meat with hydrogen gas may be considered a safe cooking method. This study could be considered a pioneer study to improve the technique of grilling meat using hydrogen gas, which is perhaps sustainable environmentally compared to other traditional gases used, such as butane. Based on this study, further research would be recommended to investigate the effect of hydrogen gas-grilling on other types of meat and with different fat levels. Further studies related to the sensory evaluation of gas-grilled horse meat using both descriptive and hedonic tests would be necessary.

## Figures and Tables

**Figure 1 foods-13-02443-f001:**
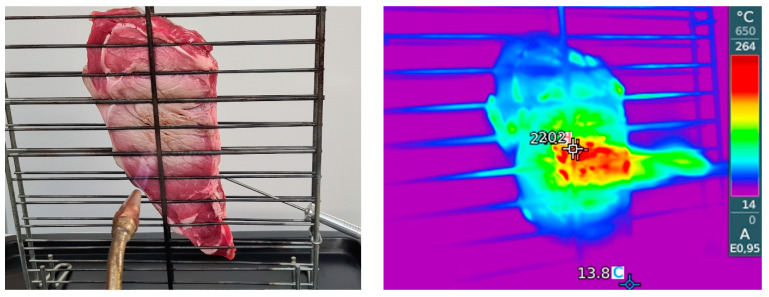
This study utilized the present assembly, which consisted of a vertically placed grill to avoid the fat dripping into the burner nozzle. Pictures in the visible (**left**) and infrared spectrum (**right**) at the start time.

**Figure 2 foods-13-02443-f002:**
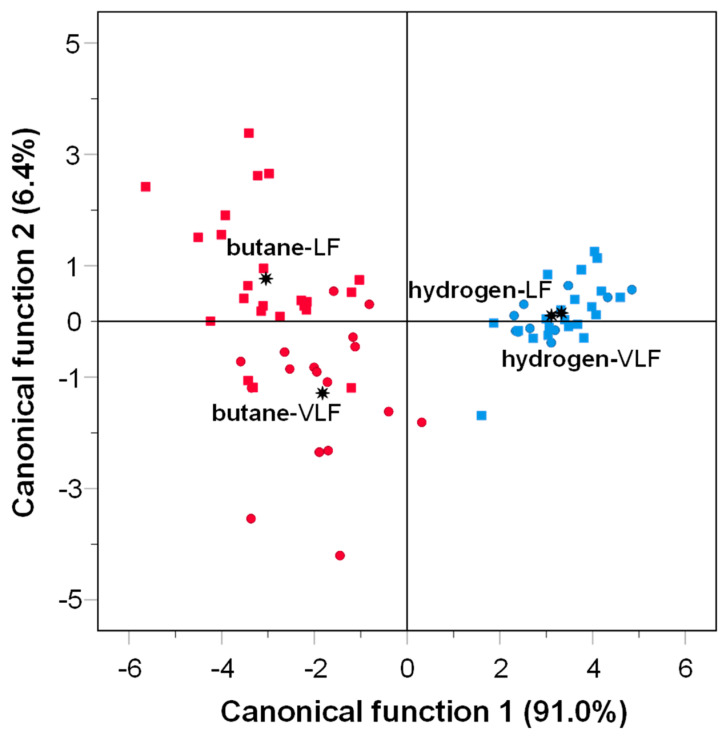
Plot of the canonical functions for the polycyclic aromatic hydrocarbons (PAHs) in horse meat samples according to fat content (●VLF: very low-fat; ■LF: low-fat) and gas-grilling (red symbols: butane and blue symbols: hydrogen). 
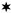
: group centroid.

**Figure 3 foods-13-02443-f003:**
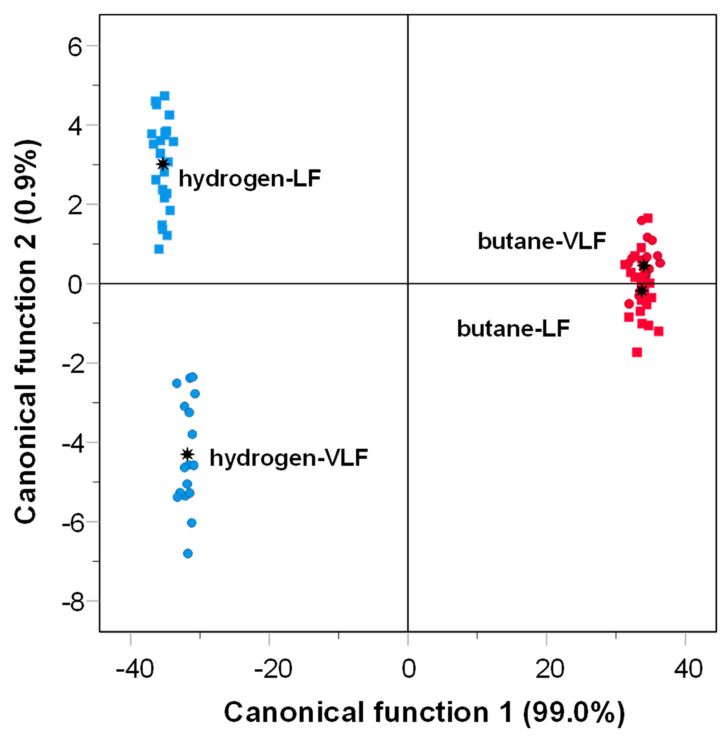
Plot of the canonical functions for the volatile organic compounds in horse meat samples according to fat content (●VLF: very low-fat; ■LF: low-fat) and gas-grilling (red symbols: butane and blue symbols: hydrogen). 
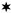
: group centroid.

**Figure 4 foods-13-02443-f004:**
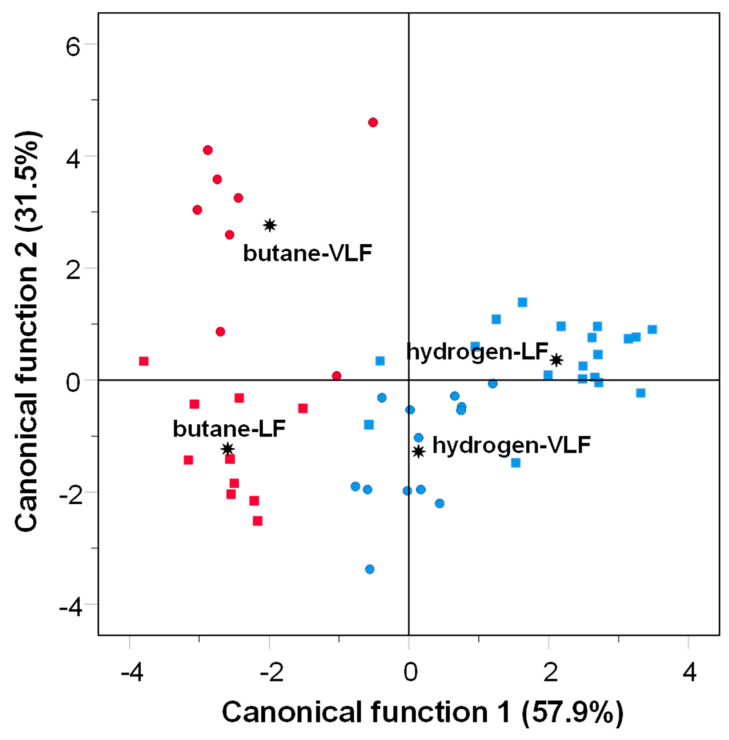
Plot of the canonical functions for the responses of analysis by e-nose in horse meat samples according to fat content (●VLF: very low-fat; ■LF: low-fat) and gas-grilling fuel (red symbols: butane and blue symbols: hydrogen gas). 
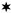
: group centroid.

**Table 1 foods-13-02443-t001:** Contents of low-molecular-weight (LMW) polycyclic aromatic hydrocarbons (PAHs) and high-molecular-weight (HMW) PAHS in horse meat samples according to fat level (very low and low) and gas-grilling (butane and hydrogen) fuel. The compounds are listed in descending order of quantity. Data (mean ± SD) are expressed as µg/kg wet weight.

	Very Low-Fat Horse Meat		Low-Fat Horse Meat	
Compounds	Butane Grill	Hydrogen Grill	Butane Grill	Hydrogen Grill
LMW PAHs				
acenaphthylene	43.45 ± 21.02 b	1.13 ± 1.06 a	73.34 ± 32.74 c	0.98 ± 0.89 a
naphthalene	41.62 ± 12.86 b	23.43 ± 3.00 a	68.63 ± 25.36 c	28.36 ± 10.73 b
pyrene	34.56 ± 19.21 b	0.55 ± 0.23 a	51.13 ± 16.74 c	0.62 ± 0.27 a
phenanthrene	26.72 ± 12.82 b	2.68 ± 0.53 a	39.01 ± 12.86 c	3.24 ± 0.65 a
fluoranthene	17.82 ± 9.87 b	0.53 ± 0.19 a	27.00 ± 8.88 c	0.59 ± 0.21 a
fluorene	7.31 ± 3.39 b	0.74 ± 0.33 a	11.15 ± 4.38 c	1.11 ± 0.33 a
acenaphthene	5.57 ± 2.14 a	7.22 ± 1.87 b	5.35 ± 1.37 a	6.06 ± 1.41 a
anthracene	4.09 ± 1.76 b	0.18 ± 0.06 a	5.76 ± 2.46 c	0.25 ± 0.08 a
chrysene	1.67 ± 0.83 b	0.17 ± 0.03 a	2.34 ± 0.79 c	0.13 ± 0.05 a
benzo[*a*]anthracene	1.60 ± 0.81 b	0.27 ± 0.09 a	2.33 ± 0.75 c	0.30 ± 0.07 a
Total LMW PAHs	184.41 ± 77.73 b	36.88 ± 5.56 a	286.03 ± 95.66 c	41.63 ± 11.47 a
HMW PAHs				
benzo[*ghi*]perylene	9.61 ± 7.09 b	0.60 ± 0.16 a	13.97 ± 4.78 c	0.41 ± 0.22 a
indeno[1,2,3-*cd*]pyrene	3.38 ± 2.23 b	0.28 ± 0.04 a	4.81 ± 1.56 c	0.24 ± 0.06 a
benzo[*a*]pyrene	2.90 ± 1.62 b	0.23 ± 0.12 a	4.46 ± 1.76 c	0.11 ± 0.07 a
benzo[*b*]fluoranthene	1.96 ± 1.73 b	0.13 ± 0.03 a	3.13 ± 1.40 c	0.19 ± 0.07 a
benzo[*k*]fluoranthene	1.06 ± 0.81 b	0.38 ± 0.08 a	1.55 ± 0.67 b	0.39 ± 0.13 a
dibenzo[*a*,*h*]anthracene	0.69 ± 0.03 b	0.38 ± 0.11 a	0.73 ± 0.08 b	0.31 ± 0.29 a
Total HMW PAHs	19.59 ± 13.36 b	1.83 ± 0.38 a	28.65 ± 9.70 c	1.61 ± 0.57 a

Different letters (a–c) in the same row are significantly different (*p* < 0.05).

**Table 2 foods-13-02443-t002:** Profile of VOCs in horse meat samples according to fat content (very low and low) and gas-grilling (butane and hydrogen). Data (mean ± SD) are expressed as AU × 10^6^/g dry matter.

	Very Low-Fat Horse Meat		Low-Fat Horse Meat	
Compounds	Butane Grill	Hydrogen Grill	Butane Grill	Hydrogen Grill
Aldehydes				
hexanal	20.03 ± 6.49 b	0.84 ± 0.44 a	21.42 ± 7.64 b	0.37 ± 0.23 a
propanal	19.75 ± 5.16	13.9 ± 3.73	18.69 ± 7.50	14.72 ± 9.86
ethanal	11.74 ± 6.58 ab	16.16 ± 17.18 b	8.74 ± 3.49 a	7.19 ± 2.09 a
pentanal	2.75 ± 1.10 b	0.08 ± 0.04 a	3.21 ± 1.48 b	0.02 ± 0.01 a
heptanal	1.57 ± 0.65 b	0.00 ± 0.00 a	1.70 ± 1.06 b	0.02 ± 0.01 a
2-methylpropanal	1.14 ± 0.44	1.56 ± 0.75	1.21 ± 0.43	1.18 ± 0.50
octanal	0.79 ± 0.36 bc	0.75 ± 0.38 bc	0.87 ± 0.42 c	0.36 ± 0.25 a
butanal	0.71 ± 0.24 a	0.75 ± 0.34 a	0.87 ± 0.22 a	1.41 ± 0.41 b
3-methylbutanal	0.60 ± 0.38 a	1.35 ± 0.79 b	0.64 ± 0.29 a	0.89 ± 0.39 a
2-methyl-butanal	0.32 ± 0.26 a	0.83 ± 0.67 b	0.28 ± 0.17 a	0.51 ± 0.28 ab
2-pentenal	0.13 ± 0.11 b	0.00 ± 0.00 a	0.19 ± 0.14 b	0.00± 0.00 a
nonanal	0.11 ± 0.11 b	0.00 ± 0.00 a	0.20 ± 0.13 b	0.00± 0.00 a
(E)-2-octenal	0.07 ± 0.05	0.20 ± 0.17	0.10 ± 0.08	0.58 ± 0.38
benzaldehyde	0.07 ± 0.05	0.27 ± 0.15	0.07 ± 0.06	0.16 ± 0.07
2-methyl-2-pentenal	<0.01 a	0.16 ± 0.13 b	<0.01 a	0.16 ± 0.13 b
Total aldehydes	59.76 ±7.92 c	36.84 ± 16.24 b	58.13 ± 9.04 c	27.37 ± 9.86 a
Alcohols				
ethanol	5.16 ± 1.45 b	1.03 ± 0.85 a	1.01 ± 0.39 a	0.63 ± 0.28 a
1-penten-3-ol	3.64 ± 1.74 b	0.04 ± 0.01 a	4.28 ± 1.58 b	0.00 ± 0.00 a
1-pentanol	0.86 ± 0.45 a	2.00 ± 1.63 b	0.80 ± 0.39 a	1.46 ± 1.14 ab
2-penten-1-ol	0.19 ± 0.16 a	0.33 ± 0.24 a	0.22 ± 0.13 a	1.12 ± 1.09 b
1-hexanol	0.04 ± 0.02 a	0.20 ± 0.10 c	0.01 ± 0.03 a	0.09 ± 0.08 b
1-octen-3-ol	0.02 ± 0.01 a	8.54 ± 4.22 c	0.01 ± 0.01 a	4.2 0± 2.35 b
1-butanol	<0.01	0.63 ± 0.43	<0.01	0.51 ± 0.28
Total alcohols	9.88 ± 3.37 ab	12.66 ± 5.57 b	6.34 ± 1.98 a	7.69 ± 3.51 a
Ketones				
2-butanone	1.11 ± 0.46	1.06 ± 0.39	0.83 ± 0.33	0.87 ± 0.38
2.3-butanedione	0.38 ± 0.26 b	0.13 ± 0.06 a	0.26 ± 0.13 b	0.13 ± 0.11 a
2-pentanone	0.20 ± 0.10 c	0.07 ± 0.06 ab	0.15 ± 0.05 bc	0.11 ± 0.10 b
2-heptanone	0.14 ± 0.08 b	0.04 ± 0.01 a	0.15 ± 0.12 b	0.03 ± 0.01 a
3-octanone	0.13 ± 0.09 a	0.61 ± 0.45 b	0.15 ± 0.08 a	1.04 ± 0.90 b
1-penten-3-one	0.11 ± 0.08 a	2.52 ± 1.08 b	0.10 ± 0.08 a	4.46 ± 1.50 c
2,3-octanediona	0.06 ± 0.03	0.07 ± 0.02	0.05 ± 0.03	0.10 ± 0.05
3-heptanone	0.03 ± 0.03 a	0.25 ± 0.10 b	0.03 ± 0.02 a	0.13 ± 0.06 b
3-hexanone	<0.01 a	0.05 ± 0.05 a	<0.01 a	0.18 ± 0.15 b
Total ketones	2.13 ± 0.77 a	4.80 ± 1.26 b	1.70 ± 0.47 a	7.05 ± 2.18 c
Aliphatic hydrocarbons				
pentane	10.72 ± 7.43	11.69 ± 5.12	9.40 ± 5.24	11.33 ± 6.76
2.2.4.6.6-pentamethylheptane	4.73 ± 2.59 b	0.09 ± 0.01 a	5.77 ± 2.95 b	0.05 ± 0.01 a
heptane	2.13 ± 1.30 a	2.49 ± 1.22 a	2.66 ± 1.23 a	4.86 ± 1.94 b
hexane	0.83 ± 0.43 a	1.02 ± 0.38 a	1.08 ± 0.37 ab	1.48 ± 0.73 b
2-pentene	0.72 ± 0.68	0.53 ± 0.48	0.68 ± 0.63	0.51 ± 0.42
3-methylene heptane	0.52 ± 0.33 b	0.04 ± 0.03 a	0.61 ± 0.37 b	0.04 ± 0.01 a
tetramethyl octane	0.34 ± 0.22 b	0.06 ± 0.05 a	0.41 ± 0.19 b	0.05 ± 0.02 a
2.4-octadiene	0.17 ± 0.08 c	0.02 ± 0.01 a	0.10 ± 0.06 b	0.02 ± 0.01 a
1,3-*trans*-5-*cis*-octatriene	0.16 ± 0.10 b	0.04 ± 0.04 a	0.22 ± 0.13 b	0.04 ± 0.02 a
2,6,7-trimethyldecane	0.14 ± 0.11 b	0.04 ± 0.02 a	0.16 ± 0.11 b	0.02 ± 0.02 a
1,3-cyclopentadiene	0.13 ± 0.10 b	0.10 ± 0.05 ab	0.14 ± 0.11 b	0.04 ± 0.01 a
3-ethyl-1.5-octadiene	0.12 ± 0.09 b	0.03 ± 0.02 a	0.17 ± 0.08 bc	0.12 ± 0.10 b
3-ethyl heptane	0.07 ± 0.06 a	0.29 ± 0.19 b	0.14 ± 0.09 ab	0.09 ± 0.05 a
decane	0.07 ± 0.05	0.26 ± 0.21	0.11 ± 0.09	0.13 ± 0.09
butyl-cyclopentane	0.06 ± 0.03 a	0.24 ± 0.16 b	0.07 ± 0.06 a	0.15 ± 0.11 ab
(*Z*)-2-octene	0.04 ± 0.01 a	13.77 ± 5.80 b	0.06 ± 0.03 a	28.71 ± 9.74 b
1-heptene	<0.01 a	0.15 ± 0.02 b	0.01 ± 0.00 a	0.13 ± 0.01 b
Total aliphatic hydrocarbons	20.81 ±9.66 a	30.78 ±10.71 b	27.77 ±8.14 a	47.75 ±9.74 c
Aromatic hydrocarbons				
benzene	1.44 ± 1.05 b	0.09 ± 0.02 a	2.08 ± 1.33 b	0.09 ± 0.06 a
toluene	1.35 ± 0.95 b	0.29 ± 0.19 a	1.29 ± 0.74 b	0.04 ± 0.03 a
ethenylbenzene	0.13 ± 0.11 a	1.66 ± 0.76 b	0.19 ± 0.09 a	2.96 ± 1.74 c
ethylbenzene	0.04 ± 0.03 ab	0.06 ± 0.02 b	0.06 ± 0.03 b	0.03 ± 0.02 a
Total aromatic hydrocarbons	2.96 ±1.93 ab	2.01 ± 0.65 a	3.63 ± 1.89 b	3.13 ± 1.73 b
Sulfur-containing compounds				
metanethiol	0.36 ± 0.18	0.27 ± 0.24	0.23 ± 0.12	0.16 ± 0.12
thiobis-methane	0.38 ± 0.28	0.31 ± 0.25	0.25 ± 0.18	0.25 ± 0.17
Total sulfur compounds	0.76 ± 0.33	0.55 ± 0.31	0.48 ± 0.26	0.41 ± 0.21
Others				
ethoxyethane	1.60 ± 1.46 ab	1.92 ± 1.15 ab	5.52 ± 4.75 b	1.09 ± 0.77 a
4-bromoheptane	0.39 ± 0.17	0.00 ± 0.00	0.26 ± 0.19	0.00 ± 0.00
2-ethylfurane	0.22 ± 0.10 a	2.73 ± 1.39 b	0.29 ± 0.24 a	3.46 ± 1.42 b
trichloromethane	0.18 ± 0.10 a	0.22 ± 0.15 a	0.49 ± 0.35 b	0.27 ± 0.18 a
hexyl formate	<0.01 a	0.13 ± 0.10 b	<0.01 a	0.14 ± 0.10 b
Total VOCs	98.48 ± 1.09	92.64 ± 14.90	97.53 ± 4.14	98.37 ± 3.15

Different letters (a–c) in the same row are significantly different (*p* < 0.05).

## Data Availability

The original contributions presented in the study are included in the article/Appendix A, further inquiries can be directed to the corresponding author.

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
