# Peer review of "Hydrogen Gas-Grilling in Meat: Impact on Odor Profile and Contents of Polycyclic Aromatic Hydrocarbons and Volatile Organic Compounds"

_foods, 2024, doi:10.3390/foods13152443_

Round 1

Reviewer 1 Report

Comments and Suggestions for Authors

To the esteemed Authors:

The manuscript need some corrections to give the high quality of a scientific report.

The detail of some corrections or errors listed below:

Line 28-29: Add reference and correct (VOCs) to (VOACs)

Line 45: Add the name of countries by reference.

Line 73: add a sentence about quantity of the consumption of horsemeat as human food in the Europe.

Line 80-84: If you compare the horizontally or vertically grilling beside other comparison, it must be add to hypothesis and goal of the research.

Line 96-98: Add a sentence (in discussion and conclusion) that if use meat by medium or highly fat content may be have different results and VOACs contents have other significant differentiation.

Line 112: ...were all in chromatography grade....

Line 115: Do you cooked each steak separately or one steak by butane and hydrogen gas? Brief it

Line 127: normally for volatile compounds must analysis directly after cooking or stored at -80 degree centigrade. Bring references for store at -20.

Line 132-145: add reference

Figure 1: why you don’t seal the body of temperature probe? As seen in the infrared spectrum photography you have temperature from torch around the body of the probe in green color. This may affect the work of probe in measuring the center temperature of the meat sample.

Line 152-153: 1mL/min? Bring reference for GC-MS conditions.

Line 154: Do the 25 °C is the elevated rate of the temperature? Correct the sentence grammatically ( ...........then...........then...........and then............??)

Line 162: it's

Line 187: Add the library update year of the MS.

Line 227-233: add statistical results between two different gas PAHs groups inside the paragraph.

Table 1: Here lower amounts of PAHs is better so the best treatment must have (a) and the others give (b, c, d) based on enhance the amount of PHAs, in statistical analysis. Correct it in all tables. Also add "Mean+-SD" in the table footnote

Line 227-228: Add the picture of the two type of the grilled meat surface or eliminate this sentence.

Figure 2: Add the symbols of VLF and LF in the legend.

Line 291-294: It is better expand this discussion based on chemical reactions.

Line 301: butane gas or hydrogen gas?

Line 313-314: I don’t believe to this conclusion. The results of the butane treatment is 4-7fold higher than hydrogen in LMW HAPs and 18-27 fold higher in HMW PAHs. Also, you used very low fat and low fat meat in this study and if you used medium or high fat horse meat the amount of the PAHs surly be higher specially in the butane treatment and it may be pass the limits of 5 and 30   µg/kg. So rewrite the sentence based on this discussion.

Table 2: comment of table 1

Line 334-337: Add discussion here that why these VOCs showed higher content in the hydrogen treatment?

Figure 3: comment of figure 2

Line 364: This is 106 or 106? Correct it in the whole manuscript

Line 365: it's

Line 420: vs. or vs.? Correct it in the whole manuscript

Line 435-438: this sentence is a repeat of a paragraph from introduction. Rewrite it or eliminate from introduction.

Figure 4: comment of figure 2. In addition, I propose use smaller size of the plot figure.

Comments on the Quality of English Language

The manuscript needs minor English editing.

Author Response

Reviewer 1

The manuscript needs some corrections to give the high quality of a scientific report.

The authors would like to thank the reviewer for their comments and suggestions for manuscript improvement. Modifications have been made to the revised manuscript with the Microsoft Word's built-in track changes function. Please find replies to each point in red on the following lines.

The detail of some corrections or errors listed below:

Line 28-29: Add reference and correct (VOCs) to (VOACs)

The reference from (Bleicher et al., 2022) has been added in lines 30-31 in the revised manuscript. To ensure uniformity in nomenclature, "volatile aromatic compounds" has been replaced with "volatile organic compounds" (VOCs),

Bleicher, J.; Ebner, E.E.; Bak, K.H. Formation and analysis of volatile and odor compounds in meat—A review. Molecules 2022, 27, 6703, doi:10.3390/molecules27196703.

Line 45: Add the name of countries by reference.

The sentence has been completed as follows (lines 46-47 in the revised manuscript):

However, these regulations allow the limits to be exceeded in some foodstuffs produced by certain countries (Ireland, Croatia, Cyprus, Spain, Poland, Portugal, Latvia, Slovakia, Finland and Sweden).

Line 73: Add a sentence about quantity of the consumption of horsemeat as human food in the Europe.

Following sentence has been added (lines 76-78 in the revised manuscript):

The European countries with the highest annual consumption of horse meat are Belgium (1.2 kg/person), Italy (1.0 kg/person), the Netherlands (1.0 kg/person) and Luxembourg (0.75 kg/person).

Weber, K.; Kearley, M.E.; Marini, A.M.; Pressman, P.; Hayes, A.W. A review of horses sent to slaughter for human consumption: impact of horsemeat consumption, residual banned drugs, and public health risks. American Journal of Veterinary Research 2023, 84, doi:10.2460/ajvr.22.10.0185.

Line 80-84: If you compare the horizontally or vertically grilling beside other comparison, it must be add to hypothesis and goal of the research

In the present study the steaks were positioned vertically to prevent the fat dripping into the burner nozzle (line 129 in the revised manuscript). No studies have evaluated the impact of gas-grilling vertically on VOCs in horse meat (line 81-82 in the revised manuscript). As reviewer indicates, we compare our results with other ones from studies in which horizontally grilling is used, because only one study (SainAubert et al., 1992) has been found that investigated vertical grilling (lines 52-54 in the revised manuscript). Thus, we have added this issue related to the position of grill in the hypothesis and goal (lines 14, 86 and 89 in the revised manuscript).

Line 96-98: Add a sentence (in discussion and conclusion) that if use meat by medium or highly fat content may be have different results and VOACs contents have other significant differentiation.

As discussed in Section 3.2, the impact of fat content on VOCs is potentially reduced by low-fat contents (lines 350-354 in original manuscript). Nevertheless, the following sentence has been added in the discussion section (lines 382-384 in the revised manuscript):

However, it should be noted that if meat with medium or high-fat content had been used, there could have been different results, and the VOCs contents might have shown another significant variation.

Line 112: ...were all in chromatography grade....

The text has been amended (line 120 in the revised manuscript).

Line 115: Do you cooked each steak separately or one steak by butane and hydrogen gas? Brief it.

One steak on each grill and with only one fuel. The clarification has been added (line 123 in the revised manuscript).

Line 127: Normally for volatile compounds must analysis directly after cooking or stored at -80 degree centigrade. Bring references for store at -20 ºC.

It can be reasonably assumed that most studies have utilized a storage temperature of –80 °C. However, in our study, we employed a storage temperature of –20 °C, in accordance with the methodology proposed by Citadini et al. (2021) (line 138 in the revised manuscript). The reference has been added.

Cittadini, A.; Domínguez, R.; Pateiro, M.; Sarriés, M.V.; Lorenzo, J.M. Fatty acid composition and volatile profile of longissimus thoracis et lumborum muscle from Burguete and Jaca Navarra foals fattened with different finishing diets. Foods 2021, 10, 2914, doi:10.3390/foods10122914.

Line 132-145: Add reference

The method reference has been included at the beginning of the sentence (line 144 in the revised manuscript) as:

The QuEChERS method adapted to meat products was used (Surma et al., 2014).

Surma, M.; Sadowska-Rociek, A.; CieÅ›lik, E. The application of d-SPE in the QuEChERS method for the determination of PAHs in food of animal origin with GC–MS detection. Eur Food Res Technol 2014, 238, 1029–1036, doi:10.1007/s00217-014-2181-4.

Figure 1: Why you don’t seal the body of temperature probe? As seen in the infrared spectrum photography you have temperature from torch around the body of the probe in green color. This may affect the work of probe in measuring the center temperature of the meat sample.

Please note that the heating process is transient and time-dependent. Figure 1 (VIS spectrum on the left and IR spectrum on the right) depicts the initial stage of the grilling process, when the meat, grill, and sensors are "cold" (green color in the infrared spectrum). Approximately 2 min of grilling are required for the meat to reach the desired temperatures. After this period, the temperatures and, consequently, the coloration in the image will be different. Figure caption has been modified as follows:

Figure 1. This study utilized present assembly, which consisted of a vertically placed grill to prevent the fat dripping into the burner nozzle. Pictures in the visible (left) and infrared (right) spectra at the start time.

Line 152-153: 1mL/min? Bring reference for GC-MS conditions.

Error has been corrected as “1 °C/min”·and method reference was Duedahl-Olesen et al. (2015) (lines 167-168 in the revised manuscript).

Duedahl-Olesen, L.; Aaslyng, M.; Meinert, L.; Christensen, T.; Jensen, A.H.; Binderup, M.-L. Polycyclic aromatic hydrocarbons (PAH) in Danish barbecued meat. Food Control 2015, 57, 169–176, doi:10.1016/j.foodcont.2015.04.012.

Line 154: Do the 25 °C is the elevated rate of the temperature? Correct the sentence grammatically (...........then...........then...........and then............??)

The sentence has been corrected as follows (lines 169-172 in the revised manuscript):

The oven program was as follows: 60 °C for one minute, then increase at 25 °C/min until reaching 180 °C, then increase at 7 °C/min until reaching 298 °C, then increase at 2 °C/min until reaching the final temperature of 325 °C and hold at that temperature for one minute.

Line 162: it's

Thank you. The error has been fixed.

Line 187: Add the library update year of the MS.

We regret to inform you that the GC-MS equipment used in the present study is not compatible with the latest software updates. The library was Wiley Volatile Compounds in Food (2018) and has been included in the revised manuscript (line 211).

Line 227-233: Add statistical results between two different gas PAHs groups inside the paragraph.

Statistical results between two different gases have been added (line 252 in the revised manuscript) as follows:

Therefore, the use of gas for grilling, compared to the use of butane, reduced the total amount of PAHs, both LMW and HMW (p < 0.05).

Table 1: Here lower amounts of PAHs is better so the best treatment must have (a) and the others give (b, c, d) based on enhance the amount of PHAs, in statistical analysis. Correct it in all tables. Also add "Mean+-SD" in the table footnote

“(mean ± SD)” has been added to the table header in all tables.

Letters for the means have been changed according to the reviewer's suggestion in all tables.

Line 227-228: Add the picture of the two type of the grilled meat surface or eliminate this sentence

We are sorry. But in the original manuscript, lines 227-228 correspond to the following sentence:

The total LMW PAHs ranged from 184.41 ± 77.73 to 286.03 ± 95.66 µg/kg in butane gas-grilled samples and 36.88 ± 5.56 to 41.63 ± 11.47 µg/kg in hydrogen gas-grilled samples.

Therefore, we don’t understand why we must delete the sentence if we don’t add the picture. Maybe this comment is related to other numbers of lines. 

Figure 2: Add the symbols of VLF and LF in the legend.

The caption now includes symbols for fat contents (circle for VLF and square for LF) and gas types (red for butane and blue for hydrogen).

Line 291-294: It is better expand this discussion based on chemical reactions.

The following explanation has been included in the revised manuscript (lines 306-321 in the revised manuscript):

Likewise, the following chemical reactions and mechanisms have been proposed for the formation of PAHs: Bitter-Howard mechanism (addition of phenyl and benzyl radicals to yield naphthalene and methylnaphthalene, respectively), Badger mechanism (dehydrogenation reactions by pyrolysis in the intermediate product generator leading to formation of 3,4-benzopyrene), Frenklach mechanism (formation of PAHS from benzene rings by a dual pathway involving phenyl and naphthalene radicals), C5H5 mechanism (decomposition of benzene to cyclopentadienyl radicals, which combine to form naphthalene), Maillard reaction mechanism (pyrolysis of the Amadori rearrangement product forms 5-hydroxymethylfurfural and furfural, both of which yield PAHs), Diels-Alder reaction pathways (cyclization reaction in which a dienophile and a conjugated diene are joined to form a cyclic product by the formation of carbon-carbon bonds), and lipid oxidation pathway (reactions in which triacylglycerides undergo decomposition, dehydrogenation, hydrolysis and oxidation processes). (Lu et al., 2024). The present study demonstrates that these reactions are modified when hydrogen is used, and further research would be interesting to understand the mechanisms of these changes when hydrogen gas is used for grilling meat.

Lu, J.; Zhang, Y.; Zhou, H.; Cai, K.; Xu, B. A review of hazards in meat products: Multiple pathways, hazards and mitigation of polycyclic aromatic hydrocarbons. Food Chemistry 2024, 445, 138718, doi:10.1016/j.foodchem.2024.138718.

Line 301: butane gas or hydrogen gas?

Production of PAH is reduced when hydrogen gas is used. The issue has been resolved (line 328 in the revised manuscript).

Line 313-314: I don’t believe to this conclusion. The results of the butane treatment is 4-7fold higher than hydrogen in LMW HAPs and 18-27 fold higher in HMW PAHs. Also, you used very low fat and low-fat meat in this study and if you used medium or high fat horse meat the amount of the PAHs surly be higher specially in the butane treatment and it may be pass the limits of 5 and 30 µg/kg. So, rewrite the sentence based on this discussion.

Thank you very much for your comment. We agree with you, and we rewrite the sentence considering your comment. The following paragraph has been added in the revised manuscript (lines 338-342):

Since detection levels were below the EC legal limits, both butane and hydrogen gas-grilling could be deemed safe from PAH contamination in horse meat with low fat content (lower than 1.9%). However, it should be highlighted than in the present study, the fat content of samples was low. Therefore, more studies with fattier meats and different species would be necessary to draw conclusions about PAH contamination in grilled meats.

Table 2: Comment of table 1

“(mean ± SD)” has been added to table header.

Letters for the means have been changed according to the reviewer's suggestion.

Line 334-337: Add discussion here that why these VOCs showed higher content in the hydrogen treatment?

Thus, a higher fat content could increase the formation of certain volatile compounds due to the thermal decomposition of lipids and their interaction with other components of the meat (Bassam et al., 2022). However, further research would be needed to precisely confirm this hypothesis for grilled horse meat cooked with hydrogen gas.

Bassam, S.M.; Noleto-Dias, C.; Farag, M.A. Dissecting grilled red and white meat flavor: Its characteristics, production mechanisms, influencing factors and chemical hazards. Food Chemistry 2022, 371, 131139, doi:10.1016/j.foodchem.2021.131139.

Figure 3: Comment of figure 2

As Figure 2, the caption now includes symbols for fat contents and gas types.

Line 364: This is 106 or 106? Correct it in the whole manuscript

Error has been fixed as 106

Line 365: it's

Thank you. Error has been fixed.

Line 420: vs. or vs.? Correct it in the whole manuscript

Please be advised that the word "vs." has been replaced by "vs." in all instances.

Line 435-438: This sentence is a repeat of a paragraph from introduction. Rewrite it or eliminate from introduction.

The sentence has been rewritten as (lines 471-474 in the revised manuscript):

The e-nose technique is not concerned with identifying and quantifying the individual components of a mixture of volatile compounds. Rather, it is a method for describing the complete aromatic profile and the interrelationships between the compounds.

Figure 4: Comment of figure 2. In addition, I propose use smaller size of the plot figure.

As figure 2, the caption now includes symbols for fat contents and gas types.

Reviewer 2 Report

Comments and Suggestions for Authors

The manuscript compares hydrogen vs. butane grilling of horse meat using laboratory method. The manuscript is composed properly and well written. The experiment design is clear and justifies the obtained conclusions. The literature review is sufficient with 35 sources mostly published within last 3 years, but also including older papers. The weakest point of the research is the equipment used for grilling. In my opinion the procedure should be described in detail (focusing also on time) and include limitations – the obtained results may not be directly reflected in traditional cooking.

Line 23 sustainability was not determine in any way in the manuscript, thus such conclusions should not be given

Line 33 fuel oil ?

Line 93-94 what was the n of LF and VLF?

Line 116-117 what was the basis for the chosen volumetric flows?

Line 469-471 indication of further research with sensory analysis could also be indicated

Author Response

Reviewer 2

The manuscript compares hydrogen vs. butane grilling of horse meat using laboratory method. The manuscript is composed properly and well written. The experiment design is clear and justifies the obtained conclusions. The literature review is sufficient with 35 sources mostly published within last 3 years, but also including older papers. The weakest point of the research is the equipment used for grilling. In my opinion the procedure should be described in detail (focusing also on time) and include limitations – the obtained results may not be directly reflected in traditional cooking.

The authors would like to thank the reviewer for their comments and suggestions for manuscript improvement. The modifications have been made to the revised manuscript with the Microsoft Word's built-in track changes function. Please find replies to each point in red on the following lines.

Line 23 Sustainability was not determined in any way in the manuscript, thus such conclusions should not be given

The authors agree with the reviewer's comment. The word “sustainability” has been deleted in the sentence.

Line 33 fuel oil ?

Thank you for pointing out the error. The word “oil” has been deleted.

Line 93-94 What was the n of LF and VLF?

A total of 60 samples of LF meat and 60 samples of VLF meat were analyzed. Sample size has been included in the text (lines 99-100 in the revised manuscript).

Line 116-117 What was the basis for the chosen volumetric flows?

In order to provide heating power at a level comparable to that of a domestic cooker burner (ranging from 0.5 to 2.0 kW), it is necessary to regulate the flow of each gas in proportion to its lower heating value (hydrogen: 120 kJ/g; butane: 45.8 kJ/g) and its molecular mass (hydrogen: 2.018 g/mol; butane: 58.120 g/mol).

Sentence has been rewritten as (lines 123-127 in the revised manuscript):

To achieve a heating power comparable to that of a domestic cooker burner (about 1.4 kW), and taking into account the lower heating values and molecular masses of the gases, the hydrogen and butane volumetric flows were 0.463 and 0.042 Nm³/h, respectively (F-112AX-HEE-99-V gas flowmeter, Bronkhorst High-Tech B.V., Ruurlo, The Netherlands).

Line 469-471 Indication of further research with sensory analysis could also be indicated

The following sentence has been added to the conclusions (lines 507-509 in the revised manuscript):

“Further studies related to the sensory evaluation of gas-grilled horse meat using both descriptive and hedonic tests would be necessary.”
